# Multi-task Learning yields Disentangled World Models: Impact and Implications

**Pantelis Vafidis**
Computation & Neural Systems
Caltech
pvafeidi@caltech.edu

**Aman Bhargava**
Computation & Neural Systems
Caltech
abhargav@caltech.edu

**Antonio Rangel**
Humanities and Social Sciences
Caltech
arangel@caltech.edu

## Abstract

Intelligent perception and interaction with the world hinges on internal representations that capture its underlying geometry ("disentangled" or "abstract" representations). The ability to form these disentangled representations from high-dimensional, noisy observations is a hallmark of intelligence, observed in both biological and artificial systems. In this opinion paper we highlight recent experimental and theoretical results guaranteeing the emergence of disentangled representations in agents that optimally solve multi-task evidence aggregation classification tasks, canonical in the cognitive neuroscience literature. The key conceptual finding is that, by producing accurate multi-task classification estimates, a system implicitly represents a set of coordinates specifying a disentangled, topology-preserving representation of the underlying latent space. Since the theory relies only on the system accurately computing the classification probabilities, we are able to derive a closed-form solution for extracting disentangled representations from any multi-task classification system. The theory provides conditions for the emergence of these representations in terms of noise, number of tasks, and evidence aggregation time, and we experimentally validate the theoretical predictions on RNNs and GPT-2 transformers solving such canonical evidence-aggregation decision-making neuroscience tasks. We find that transformers are particularly suited for disentangling representations, which might explain their unique world understanding abilities. Overall, our opinion paper puts forth parallel processing as a general principle for the formation of cognitive maps that capture the structure of the world and that are shared across both biological and artificial systems, and helps explain why ANNs often arrive at human-interpretable concepts, and how they both may acquire exceptional zero-shot generalization capabilities. We discuss implications of these findings, for machine learning and neuroscience alike.

## 1 Introduction

Humans and animals can generalize to new settings effortlessly, leveraging a combination of past experiences and world models [Lake et al., 2015, 2016]. Modern foundation models also display emergent out-of-distribution (OOD) generalization abilities, in the form of zero- or few-shot learning [Brown et al., 2020, Pham et al., 2021, Oquab et al., 2023].

Preprint.

One mechanism for generalization is through abstract, or *disentangled*, representations [Higgins et al., 2017, Kim and Mnih, 2018, Johnston and Fusi, 2023]. These two concepts are interrelated yet somewhat distinct [Ostojic and Fusi, 2024]. An abstract representation of $x_1, \ldots, x_n$ represents each $x_i$ linearly and approximately mutually orthogonally. Disentangled representations encode each $x_i$ orthogonally, without the necessity of linearity. When a representation is abstract, a linear decoder (i.e. downstream neuron) trained to discriminate between two categories can readily generalize to stimuli not observed in training, due to the structure of the representation. Furthermore, the more disentangled the representation is, the lower the interference from other variables and hence the better the performance. This corresponds to decomposing a novel stimulus into its familiar features, and performing feature-based generalization. For instance, imagine you are at a grocery store, deciding whether a fruit is ripe or not. If the brain's internal representation of food attributes (ripeness, caloric content, etc.) is disentangled, then learning to perform this task for bananas would lead to zero-shot generalization to other fruit (e.g. mangos, Figure 1a). Crucially, the visual representation of a mango is high-dimensional, non-linear and noisy, making it particularly challenging to extract a low dimensional latent like "ripeness".

Several brain areas including the amygdala, prefrontal cortex and hippocampus have been found to encode variables of interest in an abstract format [Saez et al., 2015, Bernardi et al., 2020, Boyle et al., 2022, Nogueira et al., 2023, Courellis et al., 2024]. This raises the question of under which conditions do such representations emerge in biological and artificial agents alike. Here we argue that multi-task learning is crucial to get the kind of topology-preserving representations that yield generalization in biological systems, and that a parallel processing view of the brain, in line with the cortical architecture, is naturally conducive to that framework. To do so, we first summarize findings from Vafidis et al. [2024] which proves mathematical conditions for disentanglement and experimentally confirms them in autoregressive architectures (RNNs, LSTMs, transformers) that can deal with noisy sequential real-world data, and then discuss the implications of the work for machine learning and neuroscience alike.

## 2   Problem formulation

In Vafidis et al. [2024] we are considered with canonical cognitive neuroscience tasks that involve evidence aggregation over time, mirroring decision-making under uncertainty. The tasks have a trial structure. In each trial, a ground truth vector $\mathbf{x}^* \in \mathbb{R}^D$ ($x_i^* \sim Uniform(-0.5, 0.5)$) is sampled (Figure 1b). Each element $x_i^*$ of $\mathbf{x}^*$ corresponds to different options a decision-maker might have, or to different attributes of the same item. The target output for the trial $\mathbf{y}(\mathbf{x}^*) \in \{-1, +1\}^{N_{\text{task}}}$ is a vector of $N_{\text{task}}$ +1s and -1s, depending on whether $\mathbf{x}^*$ is above or below each of $N_{\text{task}}$ classification boundaries (Figure 1b). The boundaries are fixed, and reflect criteria based on which decisions will be made. Imagine for example that $x_1$ corresponds to food and $x_2$ to water reward. Depending on the agent's internal state, one could take precedence over the other, and the degree of preference is reflected in the slope of the line.

We train RNNs and GPT-2 transformers to output the target labels $\mathbf{y}(\mathbf{x}^*)$ (Figure 1c). The networks do not have access to the ground truth $\mathbf{x}^*$ but rather a noised-up, non-linearly transformed version of it. Specifically, the input is $\mathbf{X}(t) \in \mathbb{R}^D$ where $\mathbf{X}(t) = \mathbf{x}^* + \sigma \mathcal{N}(\mathbf{0}, \mathbf{I}_D)$, $\sigma$ being the input noise standard deviation. The network should integrate noisy samples $\mathbf{X}(t)$ over time, viewed through an static, injective observation map (encoder) $f$, to estimate $\hat{\mathbf{Y}}_i(t) = \Pr\{\mathbf{y}_i(\mathbf{x}^*) = 1 | f(\mathbf{X}(1)), \ldots, f(\mathbf{X}(t))\}$.

## 3   Contributions

We here summarize the main contributions of Vafidis et al. [2024]:

- We that any optimal multi-task classifier is guaranteed to learn an abstract representation of the ground truth contained in the noisy measurements in its latent state, if the classification boundary normal vectors span the input space. Furthermore, the representations are guaranteed to be disentangled if $N_{\text{task}} \gg D$. Intriguingly, noise in the observations is necessary to guarantee the latent state would compute an optimal, disentangled representation of the ground truth (for proofs, see Appendix B of included paper).

- We confirm that RNNs trained to multitask develop abstract representations that generalize OOD as quantified by regression generalization [Johnston and Fusi, 2023] when $N_{\text{task}} \geq D$ (Figure 1f), and that the latent factors are approximately orthogonal when $N_{\text{task}} \gg D$ (disentanglement, Figure 1g). The substrate of these representations are continuous attractors [Amari, 1977] storing an estimate of $\mathbf{x}^*$ in a product space of the latent factors (Figure 1d). Furthermore, these representations preserve the topology of the real world in their structure, where a latent factor (e.g. $x_1, x_2$) corresponds to a direction in PC space of RNN hidden layer activity, and nearby trials get mapped to nearby trajectories in PC space (Figure 1d).

- We show that the setting is robust to a number of manipulations, including interleaved learning of linear and non-linear tasks and free reaction time decisions.

- We reproduce these findings in GPT-2 transformers, which generalize better due to them learning orthogonal representations for lower $N_{\text{task}}$, confirming their appropriateness for constructing world models.

- Finally, we demonstrate the strong advantage of multi-task learning, which scales linearly with $D$ and leads to representations that can be used for any task that involves the same latent variables, over previously proposed mechanisms of representation learning in the brain ("context-dependent computation") [Mante et al., 2013, Yang et al., 2019], which scale linearly with $N_{\text{task}}$ and exponentially with $D$.

Despite being framed in the context of canonical decision-making neuroscience tasks, these results are general; they apply to any system aggregating noisy evidence over time.

## 4 Implications for representation learning

**Topology-preserving representation learning**  These results have implications for the learning of representations that inherit the topological structure of the world. They suggests that this naturally happens, as long as there are enough tasks to uniquely identify the location of the ground truth $\mathbf{x}^*$ when solving these classifications (see Appendix B of Vafidis et al. [2024]). Crucially, the constraints from different tasks need to be placed simultaneously on the representation, which explains why representations emerging from context-dependent computation are typically not disentangled. An example from neurobiology is the fly head-direction system [Vafidis et al., 2022, Wilson, 2023], where a ring-like topology-preserving representation of head direction might be enforced exactly because of it's functional role in driving many downstream circuits for navigation.

**Consistency across individuals**  Potentially even more far reaching, this work implies guarantees about representational alignment across individuals or neural networks. It suggests that as long as we are faced and solve similar problems in the day-to-day world, we are bound to arrive at similar, disentangled representations of latent factors governing these decisions. This is reminiscent of the Platonic representation hypothesis [Huh et al., 2024], which suggests that the convergence in deep neural network representations is driven by a shared statistical model of reality, like Plato's concept of an ideal reality. This could explain why for example modern LLMs come to encode high-level, human-interpretable concepts [Templeton et al., 2024].

**Manifold hypothesis**  While our problem is framed in terms of arbitrary injective observation map $f$, the formulation encompasses many scenarios relevant to the manifold hypothesis [Fefferman et al., 2013]. The function $f$ can represent a smooth manifold embedded in a high dimensional space, directly modelling the manifold hypothesis of deep learning. In neuroscience, $f$ could be a non-linear encoding of stimuli in a neural population response, connecting our work to neural manifold research [Langdon et al., 2023]. By developing and testing theoretical guarantees for the emergence of disentangled representations in this multi-task problem formulation, we provide insight on how neural networks can inherently discover and linearize low-dimensional manifolds within high-dimensional, non-linear observations, enhancing our understanding of how complex data structures are captured and represented in deep learning models and biological systems alike.

**Interplay between number of tasks and fine-grainness of representations**  Intriguingly, this work reveals a fundamental interplay between richness of tasks performed and complexity/detail of the representation learned. If only a small number of tasks are performed, the resulting representations

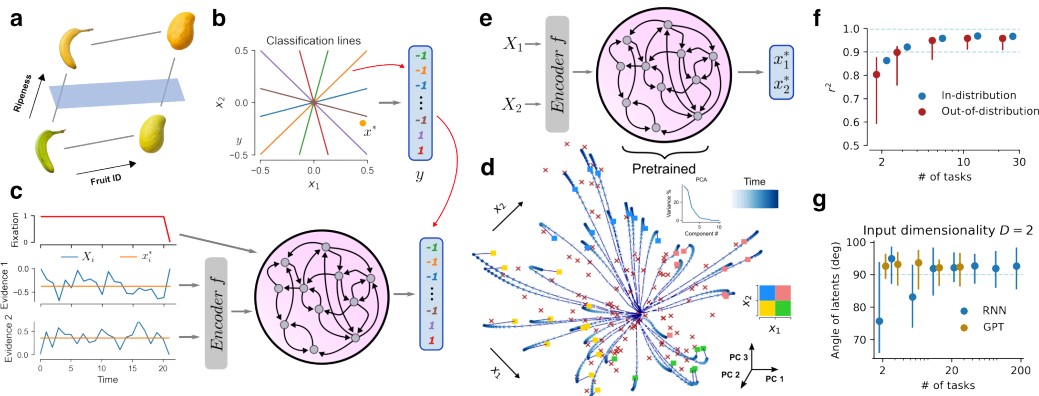

Figure 1: **Learning disentangled representations.** **(a)** A disentangled representation directly lends itself to OOD generalization: a downstream linear decoder that can differentiate ripe from unripe bananas can readily generalize to mangos, even though it has never been trained on mangos. **(b)** The task is to simultaneously report whether the ground truth $\mathbf{x}^*$ lies above ($+1$) or below ($-1$) a number of classification lines. **(c)** RNNs are trained to report the outcome of all the binary classifications in **b** at the end of the trial (indicated by the fixation input turning 0). **(d)** Top 3 PCs of RNN activity. Each line is a trial, while color saturation indicates time. All trials start from the center and move outwards, towards the location of $\mathbf{x}^*$ in state space. The last timepoint in each trial (squares) is colored according to the quadrant this trial was drawn from. Red x's correspond to attractors. Input noise here is removed so that trajectories can be visualized easier. The network learns a two-dimensional continuous attractor that seems to provide a disentangled representation of the state space. **(e)** To evaluate OOD generalization, a linear decoder (see **a**) is trained to output the ground truth $\mathbf{x}^*$ at the end of the trial, while keeping network weights frozen. The decoder is trained in 3 out of 4 quadrants and tested OOD in the 4th quadrant. **(f)** ID and OOD generalization performance for networks trained in different number of tasks $N_{task}$. The 25, 50 and 75 percentiles of $r^2$ for each network size are reported. ID and OOD performance increase with number of tasks, and the generalization gap decreases, indicating that the networks have indeed learned abstract representations. **(g)** Angles between latent factor decoders. The angles approach 90 degrees as $N_{\text{task}} \gg D$ for RNNs, but they are already close to 90 degrees for $N_{\text{task}} \geq D$ for GPT-style models. The errors that remain for $N_{\text{task}} \geq 24$ for RNNs and for $N_{\text{task}} \geq 2$ for GPT can be attributed to variability in the linear decoder fits. Therefore, we conclude that the representations become disentangled for both models. RNNs are disentangled as $N_{\text{task}} \gg D$ as our theory predicts, but GPT style models disentangle as long as $N_{\text{task}} \geq D$, showcasing their unique ability in disentangling latent factors.

will be fundamentally limited to lie within the space spanned by these tasks. However, as more tasks are added, finer details could be discerned. Therefore, the theorem and experimental results provided are not a one-way-street from dimensionality $D$ of the latent factors to how many tasks $N_{\text{task}}$ are required to uncover such latents. Rather, in a complicated and high-dimensional world, the richness of the tasks at hand directly affects the dimensionality $D$ of the latents that can be extracted, allowing for "ground truths" $\mathbf{x}^*$ at different levels of granularity to be explored. The richer the label information available, the more fine-grained the resulting world model will be.

**Disentanglement and axis-alignment**    Axis-alignment is the property by which individual neurons encode distinct latent factors, or equivalently factors are encoded across standard axis of the representation. Computer science [Higgins et al., 2017, Kim and Mnih, 2018, Chen et al., 2018, Hsu et al., 2023, Eastwood et al., 2022] and some recent neuroscience [Whittington et al., 2022] work has incorporated axis-alignment in the definition of disentanglement. However, under our definition above axis-alignment is not a requirement for disentanglement (also see Higgins et al. [2018]). Instead, we suggest that the computer science and computational neuroscience communities should adopt this broader definition of disentanglement, because otherwise we might be missing cases where the factors are not axis-aligned, but they are still orthogonal and can still be isolated by a linear decoder. Our argument is that there is nothing special about individual factors being encoded by individual neurons. Rather, we think that allowing for mixed representations within the definition

of disentanglement leads to a more holistic view of disentanglement. A contribution of our work, along with others [Johnston and Fusi, 2023], is to bring this argument to the forefront.

## 5 Connections to neuroscience and machine learning

### 5.1 Correspondence to brain processes

The brain encodes variables of interest in a disentangled format, in processes as disparate as memory [Boyle et al., 2022], emotion [Saez et al., 2015], and decision making [Bongioanni et al., 2021]. Furthermore, performance in tasks has been shown to degrade once said neural representations collapse [Saez et al., 2015], supporting the role of abstract representations in guiding generalizable behavior. Given our findings, and that the cortical architecture is uniquely suited for parallel processing [Hawkins et al., 2019], the cortex is a prime candidate area for the construction of disentangled world models. Another such area is the thalamus; it is posited that thalamocortical loops operate in parallel, and combined with internal state-dependent mechanisms lead to state-dependent action selection (e.g. prioritizing water when thirsty), while evidence integration occurs in corticostriatal circuits [Rubin et al., 2020]. The algorithmic efficiency of multi-task learning compared to alternatives ("context-dependent computation", Mante et al. [2013], Yang et al. [2019]), makes us think that it is no coincidence that the cortex is built for parallel processing; all the pieces are there, and we feel that the brain has to leverage this feature to construct faithful models of the world, as it does.

### 5.2 Multitasking vs. Multi-task learning

While our theory stems from parallel processing, i.e. multi-task learning, it is not contingent upon the parallel *execution* of multiple tasks, i.e. multitasking. Behaviorally, the agent need only perform one action, the one most appropriate to it's current internal state (e.g. thirst vs. hunger in the example above). What we posit is that tasks that have been performed by the agent before and rely on the same input are still resolved somewhere in the brain, by the brain circuits (e.g. cortical columns Hawkins et al. [2019]) previously responsible for them, instead of the entire decision-making brain area focusing only on the current task [Mante et al., 2013]. We feel that this is a more natural way of thinking about how the brain manages different tasks, with older tasks still leaving traces somewhere in the brain [Losey et al., 2024], and this theory is closely related to the widely observed phenomenon of memory replay [Foster and Wilson, 2006].

### 5.3 Relation to machine learning paradigms

The experiments in Vafidis et al. [2024] are inspired by canonical cognitive neuroscience tasks, rather than state-of-the-art ML paradigms. Yet, the conclusions concern the fundamental nature of generalization. For instance, why do foundation models generalize well in various domains? We suggest that parallel processing forces learning of generalizable world models, and our setting directly applies to settings where neural networks predict a rich representation of the world from partial observations. Some examples are predictive coding where high-dimensional next states have to be predicted [Gornet and Thomson, 2024], which is equivalent to the classification objective of predicting which objects are going to be in the field of view (and where), and self-supervised learning, where multiple missing image patches have to filled up at once [Dosovitskiy et al., 2020].

Finally, an alternative to multi-task learning that we explore is slow interleaved learning. This allows the weights of a neural network to be effectively conditioned to solve all the tasks simultaneously. The relation between multi-task and interleaved learning is a promising topic for future research.

Another interesting future direction would be to extend our work to CRNN and vision transformer architectures [Bertasius et al., 2021] that can extract latents from high-dimensional, dynamic observations e.g. video. That would require a naturalistic dataset that affords multiple views of the same object under different angles, lighting conditions etc., but still simple enough to extract useful insight. Something like a dynamic extension of dSprites [Matthey et al., 2017] would be ideally suited, however we are not currently aware of such a dataset.

## Acknowledgments and Disclosure of Funding

PV would like to thank the Onassis Foundation and AR the NOMIS Foundation for funding. AB thanks the NIH PTQN program for funding. No competing interests to declare. We would like to thank Yisong Yue for early discussions and Stefano Fusi for early feedback.

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
