# OpenReview forum: "Multi-task Learning yields Disentangled World Models: Impact and Implications"
_NeurIPS.cc/2024/Workshop/UniReps — UniReps_

### Official Review · Reviewer_Lbdb · 2024-09-28
**Review of Multi-task Learning yields Disentangled World Models: Impact and Implications**

**Rating:** 6
**Confidence:** 1

**Review:**

Quality:
The work provides a detailed exploration of the emergence of disentangled representations in multi-task learning, backed by theoretical results and experimental validation.

Clarity:
The paper is well-structured and presents its findings in a clear and organized manner. It effectively communicates complex concepts related to disentangled representations, multi-task learning, and their implications for both machine learning and neuroscience. The figure of learning disentangled representations helps readers grasp the core concept of paper quickly.

Originality:
The work showcases originality by highlighting the emergence of disentangled representations in agents solving multi-task evidence aggregation and classification tasks. It also emphasizes the role of parallel processing in constructing cognitive maps shared across biological and artificial systems, contributing to the originality of the findings.

Significance:
The work holds significant implications for both machine learning and neuroscience. It sheds light on the generalization abilities of biological and artificial systems, as well as their exceptional zero-shot generalization capabilities. The implications for representation learning and topology-preserving representation learning are also noteworthy. The findings that transformers are particularly suited for disentangling representations could lead to better explanation of unique world.

Pros:

1. Clear and organized presentation of findings

2. Original insights into disentangled representations and multi-task learning

Cons:

1. Need more explicit discussion of the potential practical applications of the findings.

---

### Official Review · Reviewer_aw1L · 2024-09-29
**Interesting findings about the impact of multi-task learning on the learned latent space**

**Rating:** 6
**Confidence:** 3

**Review:**

Summary:
The work analyzes how training RNN models and GPT-2 models on multiple tasks ($N_{tasks}$) can cause disentanglement of the latent space of dimension ($D$). The authors discuss and conclude that when $N_{tasks} \gg D$, the disentanglement is pronounced. This entanglement is further correlated with OOD generalization of the trained model through an independent series of experiments.

Pros:
1. The paper provides rigorous theoretical guarantees for disentangled representations in multi-task classification
2. The authors validate their theoretical results across various neural architectures (RNNs, transformers) and tasks. The experiments demonstrate that abstract, disentangled representations lead to zero-shot out-of-distribution (OOD) generalization​
3. The paper connects neuroscience and machine learning by drawing parallels between cognitive processes (e.g., decision-making) and neural networks. This link could be insightful for both fields
4. The paper comprehensively discusses connections to existing theories in neuroscience and AI, making the findings applicable to both biological and artificial systems

Cons:
1. While the paper presents detailed experiments, most are neuroscience-based. It lacks results on standard ML benchmarks, such as dSprites, which could have made the results more directly comparable to existing methods.
2. The findings indicate that noise is essential for achieving disentangled representations, but do not address limitations associated with having noise in the labels, such as slow convergence and instability of training.
3. Search space is limited to RNNs and GPT-2-like models to perform classification tasks and the extension of the findings to other types of neural networks such as CNNs is not tested or discussed, limiting the direct extension to vision-related applications

---

> ### Author Response · Authors · 2024-10-16
> **Reply to reviewer**
>
> We would like to thank the reviewer for their detailed comments. In regards to the specific points raised:
>
> 1. While we agree that our study focuses on neuroscience tasks, as we emphasize in the limitations in the highlighted work there are no ML benchmarks such as dSprites for the kind of dynamical, noisy tasks that we are dealing with here. Future work could address this shortcoming by creating new datasets that extend dSprites in more dynamic, real world settings e.g. where items are viewed from different angles and light conditions in a grocery store. An alternative would be to apply our framework to online retailer data where sequential decisions like the ones we study are made; however no datasets of this nature are publicly available to our knowledge.
>
> 2. While noise is bound to slow down the pre-training process, it is a limitation that agents operating in the real world would have to face anyway. In terms of the post-training usage of the representation by linear decoders, this is a convex optimization problem with convergence guarantees.
>
> 3. While we agree that an extension to vision architectures would be interesting, that would require a considerable overhaul of our approach including data and architecture, where more naturalistic data and deep CRNN-types of architectures might have to be used. For this reason, this would constitute an entire new study on its own, and future work could endeavor to do this. In the noise-free regime on the other hand, things are more straightforward and models that are using convolutional feedforward architectures have been shown to disentangle (Maziarka et al 2022).

---

### Official Review · Reviewer_Jw8F · 2024-10-06
**Revision of UniReps '24 — #38**

**Rating:** 5
**Confidence:** 4

**Review:**

**Paper summary**

In this paper, the authors argue that disentangled representations emerge in agents that optimally solve multi-task evidence aggregation classification tasks, canonical in the cognitive neuroscience literature. The key conceptual finding is that, by producing accurate multi-task classification estimates, a system implicitly represents a set of coordinates specifying a disentangled, topology-preserving representation of the underlying latent space. The proposed theory provides conditions for the emergence of these representations in terms of noise, number of tasks, and evidence aggregation time, and they experimentally validate the theoretical predictions on RNNs and GPT-2 models solving such canonical evidence-aggregation decision-making neuroscience tasks.

**Paper strengths**

The argument posed in this paper is interesting, and the application in the neuroscience research field is quite novel.
The authors present results using both traditional RNNs and more recent models like GPT-2.

**Paper weaknesses**

This paper is a condensed version of the full article, "Disentangling Representations through Multi-task Learning", which is provided as supplementary material. Therefore, it does not contain any significant improvements/new claims compared to the extended version, but is only a summary of it.

**Suggestions**

To expand and improve this work further, the authors should check the literature in other research fields in which several works have already been presented regarding the relationship between disentanglement and MTL.
For example:
* Meng et al. [1] highlight a link between disentangled representations and MTL, demonstrating that disentangled features can enhance multi-task network performance, particularly on datasets with novel properties.
* Yang et al. [2] introduce the concept of "Knowledge Factorization", where the knowledge from a pre-trained multi-task network (the teacher) is leveraged to train disentangled single-task networks (the students), optimizing computational efficiency for the final single-task network. Their factorization approach is twofold: structural factorization, which separates the network into a shared knowledge network and a task-specific network, and representation factorization, which is driven by mutual information.
* Maziarka et al. [3] conducted a disentanglement analysis of MTL models using a semi-synthetic dataset derived from latent information in simple datasets.
* Skenderi et al. [4] demonstrate that disentangling the representation space can be a general prior for MTL. By leveraging disentanglement to identify auxiliary tasks, an MTL model can extract a model-specific embedding that incorporates both the primary task and the newly discovered labels, ultimately enhancing performance on the primary task.

**References**

[1] Meng, Qingjie, et al. "Representation disentanglement for multi-task learning with application to fetal ultrasound". Smart Ultrasound Imaging and Perinatal, Preterm and Paediatric Image Analysis: First International Workshop, SUSI 2019, and 4th International Workshop, PIPPI 2019, Held in Conjunction with MICCAI 2019, Shenzhen, China, October 13 and 17, 2019, Proceedings 4. Springer International Publishing, 2019.

[2] Yang, Xingyi, Jingwen Ye, and Xinchao Wang. "Factorizing knowledge in neural networks". European Conference on Computer Vision. Cham: Springer Nature Switzerland, 2022.

[3] Maziarka, Łukasz, et al. "On the relationship between disentanglement and multi-task learning". Joint European Conference on Machine Learning and Knowledge Discovery in Databases. Cham: Springer International Publishing, 2022.

[4] Skenderi, Geri, et al. "Disentangled Latent Spaces Facilitate Data-Driven Auxiliary Learning". arXiv preprint arXiv:2310.09278 (2023).

---

> ### Author Response · Authors · 2024-10-16
> **Reply to reviewer**
>
> We would like to thank the reviewer for their comments, and for pointing these references to us.
>
> Most of these references are on disentanglement in feedforward architectures, and we focus in recurrent architectures; yet we found [3] particularly useful since it seems that its approach and results are very similar to Johnston & Fusi (2023) which we discuss extensively, and while we focus on the Multi-task Learning -> Disentanglement connection [4] shows that the opposite direction is also possible. Therefore, we plan to include these references in future versions of the highlighted work where we discuss connections to the literature in detail.
>
> On that note, we would like to remind the reviewer that these results have not been presented before, as this constitutes an opinion paper based on unpublished results.

---

### Official Review · Reviewer_3GBZ · 2024-10-06
**Good paper very relevant to the audience**

**Rating:** 10
**Confidence:** 4

**Review:**

In this work, the author’s present an insightful hypothesis: multitasking (parallel processing). Specifically, that abstract representations result from the need to use the same feature representations to solve a multitude of tasks. Following from this hypothesis, the author’s derive conditions that guarantee the emergence of abstract representations: namely the number of simultaneous tasks being performed and show that their theory holds up in three classes of sequence models RNNs, LSTMs and Transformers.

I think this work is quite strong and the results are both clear and impactful. As the author’s mentioned in this paper, their theory provides guarantees on when two people, or a model and a person will form the same representations. This alone is an important finding that already suggests a myriad of future experiments. Yet, the author’s also point out an equally interesting observation: that is that Transformers seem to have an implicit bias leading them to naturally develop abstract representations at least in comparison to RNNs. Understanding the source of this bias (is it architectural? Is it due to the attention mechanism?) will be particularly insightful for developing learning theories and broadly interesting to the machine learning community as a whole.

That all of these results stem from the assumption that brain or the model must simultaneously perform multiple tasks is interesting and has broad implications to neuroscience. However, I think more could be done to strengthen this connection. Namely, a key assumption for this theory is the simultaneity of tasks. Many experimental tasks are sequential – sometimes interleaved or blocked. In the full paper, the author’s show that their theory generalizes to the interleaved training regime, but in my experience blocked training is much easier for animals and probably humans. Relatedly, work from Anne Collin’s group has shown that whether animals are trained with interleaved or blocked trial designs has strong implications for which strategies they develop and the generalization properties of those strategies. I would be curious to see how the blocked training regime can be encapsulated in this model and what kind of results emerge. Another key assumption is that the representations are disentangled. Can the authors say anything more about which tasks this is likely to hold for (e.g. sensory discrimination tasks but not motor learning tasks)? Could these results be generalized and extended to meta-learning as well? For instance, does training networks to perform the Harlo task bias them towards learning abstract representations in future tasks?

But these are just more examples of interesting questions spawned by this work.

All said, this is an excellent and thought provoking paper, and anyone interested in how the brain or neural networks in general develop their representations, or which representations lead to generalization would benefit from hearing about this work.

---

> ### Author Response · Authors · 2024-10-15
> **Reply to reviewer**
>
> We thank the reviewer for their in depth comments and for acknowledging the implications of this work. In regards to the specific comments:
>
> ### Motor learning
>
> As an output-driven process, motor learning involves the learning of highly specialized trajectories that are specific to the task being executed. We could, however, see our framework leading to the learning of motor primitives, if several such motor executions are vicariously planned in parallel (efferent copy etc.). Then the motor primitives could be combined to form motor plans, the same way features are combined to form objects.
>
> ### Meta-learning
>
> Our work can be though as already implementing meta learning, since the learned representation can be used for *any* downstream task involving the same latent variables.
>
> ### Interleaved and blocked training
>
> The reviewer makes an insightful observation that humans and animals excel at blocked learning. On the other hand, neural networks generally perform best at interleaved learning, which allows their weights to be gradually conditioned to solve all tasks. Some recent work has attempted to bridge the two (Flesch, T., Nagy, D. G., Saxe, A., & Summerfield, C. (2023). Modelling continual learning in humans with Hebbian context gating and exponentially decaying task signals. PLoS computational biology, 19(1)), but the setting is quite simplified. Our intuition is that this difference in performance across different learning regimes rests upon high-level learning biases that living organisms have, related to their allocation of attention etc., for which phenomena neural networks might not be the most appropriate level of explanation. Yet, we think that relaxing the simultaneity assumption in our work for interleaved and even blocked training settings would be a very promising area for future work.

---

### Decision · Program_Chairs · 2024-10-10

**Decision:**

Accept

**Comment:**

In light of the positive reviewers' feedback and relevancy of the submission, we are pleased to accept this paper for presentation at UniReps 2024. We kindly ask the authors to incorporate the reviewers' suggestions and feedback in the final camera-ready version of the manuscript.